# A new regionally consistent exposure database for Central Asia: population and residential buildings

Chiara Scaini[1], Alberto Tamaro[1], Baurzhan Adilkhan[2], Satbek Sarzhanov[2], Vakhitkhan Ismailov[3], Ruslan Umaraliev[4], Mustafo Safarov[5], Vladimir Belikov[6], Japar Karayev[6], Ettore Faga[7].

[1]National Institute of Oceanography and Applied Geophysics – OGS, Trieste, 34100, Italy
[2]Institute of Seismology, Ministry of Emergency Situations, Almaty, 050060/A15E3F9, Kazakhstan
[3]Institute of Seismology of Uzbekistan, Tashkent, 700128, Uzbekistan
[4]Institute of Seismology of Kyrgyz Republic, Bishkek, 720060, Kyrgyzstan
[5]Research Center for Ecology and Environment of Central Asia, Dushanbe, 734063, Tajikistan
[6]Independent consultant, Turkmenistan
[7]RED Risk Engineering Development, Pavia, 27100, Italy

*Correspondence to*: Chiara Scaini (cscaini@ogs.it)

**Abstract**
Central Asia is highly exposed to a broad range of hazardous phenomena including earthquakes, floods and landslides, which have cause substantial damages in the past. However, disaster risk reduction strategies are still under development in the area. We provide a regional-scale exposure database for population and residential buildings based on existing information from previous exposure development efforts at regional and national scale. Such datasets are complemented with country-based data (e.g. building census, national statistics) collected by national representatives in each Central Asia country (Kazakhstan, Kyrgyz Republic, Tajikistan, Turkmenistan, Uzbekistan). We also develop population and residential buildings exposure layers for the year 2080, which support the definition of disaster risk reduction strategies in the region.

**Short summary (plain text)**
Central Asia is highly exposed to multiple hazards, including earthquakes, floods and landslides, for which  risk reduction strategies are currently under development. We provide a regional-scale database of assets at risk, including population and residential buildings, based on existing information and recent data collected for each Central Asia country. Population and number of buildings are also estimated for the year 2080 to support the definition of disaster risk reduction strategies.

## 1. Introduction

Central Asia is highly exposed to a broad range of hazardous phenomena including earthquakes, floods and landslides. Such disasters can affect single countries but often have trans-boundary consequences. In addition, disaster risk and subsequent losses are expected to increase under the effect of climate change (Yuyu et al., 2019). For these reasons, a regional-scale approach is needed to support, plan and coordinate Disaster Risk Reduction (DRR) strategies in the Central Asia region. Such approach should rely on evidence-based technical and scientific assessments of all elements that concur to risk. In particular, exposure plays a paramount role in disaster risk reduction by supporting the identification of the number and type of assets damaged or disrupted by hazardous phenomena (Pittore et al., 2017). For DRR purposes, it is particularly relevant to know the number and characteristics (e.g. demographics) of occupants to define mitigation measures (e.g. evacuation plans) and long-term preparedness programs (e.g. education activities). Knowledge on the typology and characteristics of residential buildings is also paramount in order to assess which buildings can suffer damages and the potentially harmed or

stranded occupants. Finally, exposure layers provide a financial indicator on the exposed assets value, in particular buildings,
to support regional disaster risk reduction and financial risk mitigation activities.

In Central Asia, strong efforts were devoted to assessing expected hazard and to estimating risk for specific hazardous phenomena (e.g. earthquakes). However, most risk assessment efforts were focused on single countries and hazards, such as during the project "Measuring Seismic Risk in Kyrgyz Republic", developed by World Bank in the period 2014-2017. During the EMCA project (Earthquake Model Central Asia, https://www.emca-gem.org/), a first important step was taken
towards unifying hazard, exposure, vulnerability and risk assessment at the regional scale for Central Asia. However, the effort was focused on seismic risk, while less attention was devoted to assessing impacts of other hazardous (floods, landslides) at the regional scale. Flood hazard, nonetheless, has become increasingly relevant in Central Asia causing impacts that were often exacerbated by the difficulties of trans-boundary cooperation (e.g. concerning reservoirs' operation and maintenance, UNECE 2011; Libert and Trombitcaia, 2015). Following earthquakes and floods, landslides are the third
most prevalent natural hazard in Central Asian (CACDRMI, 2009) and are often triggered by natural events such as earthquakes, floods, rainfall and snowmelt (Saponaro et al., 2014; Strom and Abdrakhmatov, 2017). The population of Central Asia is steadily growing and is expected to exceed the 100 million people by 2050, with a much higher growth rate than the world average (36.9% against 26.2%, https://www.eurasian-research.org/publication/un-population-prospects-case-of-central-asia/). The most populated country are Uzbekistan and Kazakhstan but population density is unevenly distributed
in the region with almost 50% of the population concentrated in few densely populated cities (Seitz, 2019). Given the the wide range of impacts that might be caused by earthquakes, floods and landslides and their potential interaction beyond country boundaries, a regional-scale exposure database is nowadays of paramount importance. The only regional-scale exposure datasets of residential buildings available at the time (April 2023) are provided by Pittore et al., (2020) and Yepes-Estrada et al. (2023). The dataset provided by Pittore et al. (2020) relies on ground-based and remote sensing data in Kyrgyz
Republic and Tajikistan (Wieland et al., 2012; 2015) and was designed for the purpose of seismic risk assessment and has a variable spatial resolution, obtained by Voronoi tesselation, which is coarser in rural areas. Similarly, the database provided by Yepes-Estrada et al. (2023), which also assimilates the dataset of Pittore et al., (2020), makes use of bottom-up approaches to produce an updated exposure layer at a resolution of approximately 30km. Replacement costs provided by Pittore et al., (2020) were derived based on costs obtained from specific studies developed on Kyrgyz republic (Arup, 2016),
but required additional validation based on more recent country-based data for all 5 countries of Central Asia. For these reasons, a regionally-consistent exposure dataset with latest information on population, residential buildings and associated replacement costs obtained from local representatives was needed. In addition, for the purpose of flood and landslide risk assessment, the resolution had to be increased. In this study, we assembled the first high-resolution (500m) regionally consistent exposure database of population and residential buildings exposed to earthquakes, landslides and floods in Central
Asia. The dataset was developed using the last available census of population and buildings and recent construction costs provided by local partners of the consortium in each of the 5 countries of Central Asia.

Exposure databases do not only support current risk assessment estimates, but can inform strategies for the mitigation of future risks, which might be exacerbated by long-term phenomena (e.g. climate change). This requires projecting the exposure to represent the future situation, e.g. at the end of the century. At the time, no future dataset of population and
residential buildings are currently available for Central Asia. Shared Socio-economic Pathways (O'Neill et al., 2014) represent possible developments scenarios over a century timescale based on different economic, environmental and social policies. Here, we present the first exposure dataset for 2080 developed for three selected SSPs in order to support the definition of long-term disaster risk reduction strategies at the regional scale. Future urban area layers were developed at global scale for different SSPs developed specifically for Central Asia (Pedde et al., 2019). The work was developed within
the SFRARR program ("Strengthening Financial Resilience and Accelerating Risk Reduction in Central Asia"), promoted by European Union, aims at leveraging all risk-related data and assessments in order to quantify financial disaster risk in

Central Asia. The program focused on earthquakes, floods and landslides, and envisaged the creation of the first high-resolution (500m) regionally-consistent exposure database for multiple hazards in Central Asia.

**2. Data collection**

The regional-scale exposure layers for Central Asia were developed based on data collected at two spatial scales: Global/regional and national/sub-national.

- Global/regional. Global and regional-scale data were collected from existing official sources and literature works, following the suggestions of international experts in the region, such as the Regional Scientific-Technical Council (RSTC), constituted in the framework of the EU SFRARR Program. In general, these databases have a large coverage, but often with lower spatial resolution. For the development of population exposure layers, the Facebook global dataset was retrieved for the year 2020, available at the Humanitarian Data Exchange webpage (https://data.humdata.org/organization/facebook). It contains the total population at 30-m resolution and the fraction of population by gender and age classes. As for residential buildings, the regional-scale layer of Pittore et al. (2020) is the most recent available exposure database for the region. The spatial distribution of urban and rural areas was retrieved from the Global Human Settlement Layers (GHSL) (JRC, 2021) at 1km resolution for the years 2000 and 2015. Spatial layers of expected urban area in 2080 under different SSPs were provided Chen et al. (2020) at 1 km resolution.

- National/sub-national scale. The data collection was performed by the exposure working group, constituted by contact persons for each of the 5 countries of Central Asia who collected data both from national ministries (e.g., census data) and from past projects carried out in their country. Local partners collected the population census for the latest year available (2021 for Uzbekistan, 2020 for Kazakhstan and Kyrgyz Republic, 2019 for Turkmenistan, and 2018 for Tajikistan). For two countries, Kazakhstan and Uzbekistan, information about the number of households by Oblast and load-bearing material was available. National and sub-national official data are usually provided by recognized institutions (e.g. national ministries) and have higher spatial resolution with respect to global or regional data. However, their availability is limited to some countries, such in the case of building census. In addition, local experts can provide additional data related to their judgment (e.g. expert opinion) which support the exposure development.

**3. Methodology**

The exposure assessment is based on the combination of data collected at two spatial scales: global/regional and national/sub-national. The underlying assumption is that recent country-based data (national or sub-national scale) are more reliable than global or regional layers. Based on these considerations, existing global/regional layers were complemented with national or sub-national scale, as described in the following subsections for population and residential buildings.

**3.1 Development of population exposure layers**

The population exposure layer was developed based on the Facebook high-resolution dataset (https://data.humdata.org/), which was enhanced using the country-based demographic information. Population data in the Facebook dataset, originally available at 20m resolution, was aggregated at 100m resolution and classified into three age intervals: population younger than 5 years old, older than 60 years old or in the intermediate age class. The Facebook data was then compared with national census data collected by local partners. This includes population data by age and gender in each country and sub-national administrative units (*oblasts*) extracted from the latest available national census (2021 for Uzbekistan, 2020 for Kazakhstan and Kyrgyz Republic, 2019 for Turkmenistan, and 2018 for Tajikistan). Differences on the total population

exceeded the 20% in 7 o*blasts* over a total of 44 oblasts considered in the study. The comparison showed that at the regional scale, the Facebook dataset contains a 5% less population with respect to the national census retrieved. At the national scale, the population in the Facebook dataset was also consistently lower than in national census, with a difference of 1.5, 4, 5 and 8% respectively for Kazakhstan, Uzbekistan, Turkmenistan and Kyrgyz Republic. We noticed that larger discrepancies were associated with the presence of older census data (e.g. for Turkmenistan) while smaller differences are found in Kazakhstan, and Uzbekistan, with the exception of Kyrgyz Republic where discrepancies were high despite the census was relatively recent. The estimated difference was then used to refine the Facebook dataset, under the assumption that updated country-based data are more reliable than regional datasets. For Tajikistan, given that the retrieved data was older than the Facebook dataset and was available only for selected towns, so we did not correct the Facebook layer assuming it to be more reliable. The correction was performed on a cell-by-cell basis, and proportionally to the estimated difference between the two datasets. The same procedure was applied at the city scale to a number of cities in Kyrgyz Republic, Kazakhstan and Turkmenistan, for which population data were available. Gender and age percentages were also corrected with the exception of the elder fraction because the data at national scale was only available for different age thresholds (e.g., 70 for Kyrgyz Republic and Uzbekistan, 63 for Kazakhstan). The population layer was validated using data collected by local partners for specific cities. The final dataset was produced at a resolution of 100m.

## 3.2 Development of residential buildings exposure layers

The exposure assessment of residential buildings consists in defining dominant building typologies (codified by taxonomies). For residential buildings, pre-defined typologies were available from a previous project EMCA (The Earthquake Model Central Asia). Typologies were defined based on national-scale surveys in particular in Kyrgyz Republic and Tajikistan and extended to the entire Central Asia region (Wieland et al., 2015). EMCA typologies are described based on the Global Earthquake Model (GEM) building taxonomy (Silva et al., 2022) In this work we updated the existing typologies with the information collected at national scale by local partners and their associated taxonomy (Table 1). In particular, country-based census data for 2020 were collected for Kazakhstan and Uzbekistan, for which the building census provided the number of buildings per typology aggregated at the *oblast* level. A correspondence was defined between the national census typologies and the ones in the EMCA classification based on the typologies description, pictures and input provided by local partners.

**Table 1:** Building typologies defined for residential buildings in Central Asia, based on the previous work of Wieland et al., (2015) and Pittore et al., (2020). Each EMCA typology and sub-typology is associated with age and storey number (expressed in ranges), average floor area, number of households and average occupancy. The taxonomy in the GEM format (Silva et al., 2022) is also provided.

| Typology | Sub-typology | Age | Storey number | Floor area (m²) | House-holds | Average occupancy | Taxonomy |
|---|---|---|---|---|---|---|---|
| EMCA1 | URM1 | 1930-1960 | 2-4 | 250 | | 3.8 | /MUR + CLBRS + MOC/LWAL + DNO/FW + HBET:2,4 + YBET/1930,1960 |
| | URM2 | | 1-2 | 150 | 1 | | MUR+ MOCL/LWAL + DNO/FC + HBET:1,2 + YBET/1930,1960 |
| | CM | 1960-2001 | 1-5 | 2000 | | 76 | /MCF + MOC/LWAL + DNO/FC/HBET:1,5 + YBET/1960,2001 |
| | RM-L | | 1-2 | 250 | | 5.2 | /MR + MOC/LWAL + DNO/FC/HBET:1,1 + YBET:1960,2001 |
| | RM-M | | 3-4 | 2000 | 12 | 104 | /MR + MOC/LWAL + DNO/FC/HBET:3,4 + YBET:1960,2001 |
| EMCA2 | RC1 | 1957-2006 | 3-7 | 1500 | 45 | 152 | /CR + CIP/LFM + DUC/FC/HBET:3,7 + YBET:1957,2006 |

| | | | | | | | |
|---|---|---|---|---|---|---|---|
| | RC2 | 1957-2021 | 4-9 | 2000 | | 190 | /CR + CIP/LDUAL + DNO/FC/HBET:4,9 + YBET:1957,2021 |
| | RC3 | 1957-2021 | 2-5 | 1500 | | 152 | /CR + CIP/LFINF + DNO/FC/HBET:2,5 + YBET:1957,2021 |
| | RC4 | 1957-2006 | 4-16 | 5000 | | 190 | /CR + CIP/LWAL +DNO/FC/HBET:4,16 + YBET:1957,2006 |
| EMCA3 | RCPC1 | 1956-1980 | 1-16 | 5000 | | 152 | /CR + PC/LWAL + DUC/FC/HBET:1,16 + YBET:1956,1980 |
| | RCPC2 | 1980-2021 | 3-12 | 5000 | 70 | | /CR + PC/LFLS + DUC/FC/HBET:3,12 + YBET:1980,2021 |
| EMCA4 | ADO | n.a. | 1 | 100 | 1 | 5.2 | /MUR + ADO/LWAL + DNO/FW/HBET:1 |
| EMCA5 | WOOD1 | to present | 1-2 | 150 | | 3.8 | /W/LWAL + DUC/FW/HBET:1,2 + YPRE:2021 |
| | WOOD2 | <1980 | 1-2 | 150 | 1 | | /W+ WLI/LO + DUC/FW/HBET:1 |
| EMCA6 | STEEL | n.a. | 1 | 2000 | 1 | 3.8 | /S/LFM +DNO/FME/HBET:1 |

The spatial distribution of building typologies was derived from the layer provided by Pittore et al., (2020) which collects all the previous information generated by the EMCA project. The original layer has a variable resolution ranging from a few hundred meters in urban areas to several km in rural areas and was developed specifically for earthquake risk assessment purposes. First, the layer of Pittore et al. (2020) was updated using recent census data at the *oblast* level, available for Kazakhstan and Uzbekistan. Then, the spatial resolution was increased to produce a residential buildings exposure layer on a 500-m-resolution grid and support risk assessment for flood hazard. The procedure, exemplified in Fig. 1 for Kazakhstan. comprised three main steps:

- Each cell of the variable-resolution layer was identified as urban or rural based on the 2015 Global Human Settlement Layers layer (JRC, 2021). GHSL cells associated with a city code were classified as urban, while urbanized areas without city code, which correspond to villages, were assumed to be rural. Each urban/rural area was associated with a distribution of EMCA building typologies, provided by Wieland et al. (2015) for each Central Asia country.
- The collected country-based information for Kazakhstan and Uzbekistan was integrated in the layer of Pittore et al. (2020) maintaining its original spatial resolution. The number of buildings in each EMCA typology and in each Oblast was distributed on the variable-resolution grid, using the total population in each cell (included in the layer of Pittore et al., 2020) as a proxy of their spatial distribution. The procedure was performed for each EMCA typology and accounted for the different building types distribution in urban and rural areas identified in the previous step.
- Distribution of residential buildings on a regular grid of 500m. This was done in two steps: first, the buildings in each variable-resolution cell were distributed on on a new, regular grid of 30-m resolution based on the Facebook population layer. Population density in the 30-m layer was therefore used as a proxy of the buildings presence. The final residential buildings exposure layer was assembled by aggregating the values at 500-m resolution. A few simple checks were performed in order to make sure that no points were associated with null population and not-null number of buildings, or vice-versa, and that the average number of occupants per building was consistent with the average occupancy defined for Central Asia typologies (Table 1).

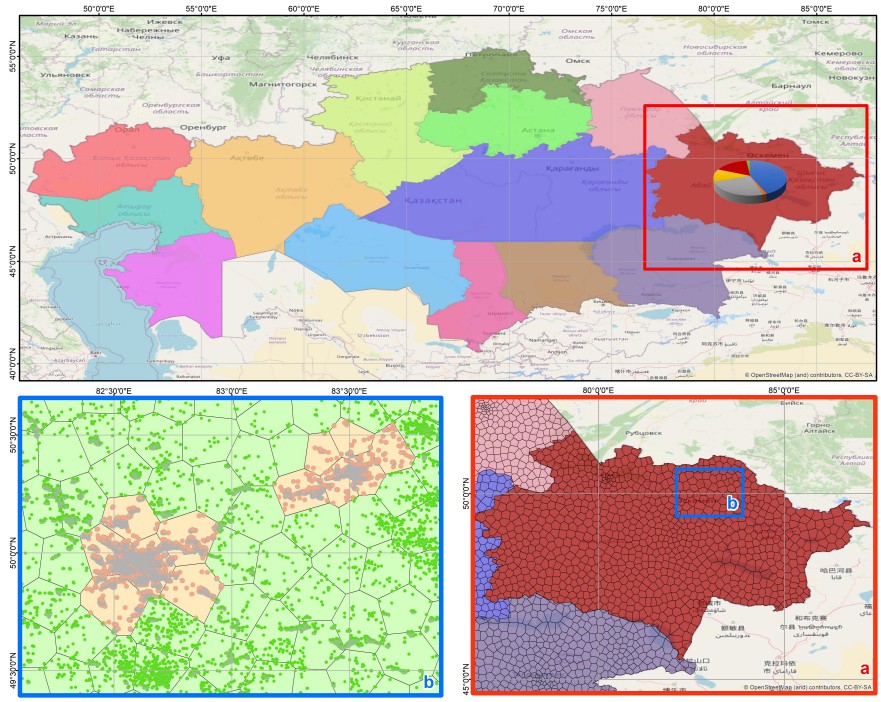

190

**Fig.1: Methodology for obtaining the exposure layer of residential buildings at 500-m resolution, exemplified for Kazakhstan.** First, the existing exposure layer of Pittore et al., (2020) is integrated with country-based information provided by local partners for each Oblast (a). Secondly, the information is disaggregated on a regular 500-m grid (b). The different distribution of building typologies in urban and rural areas was accounted for following Wieland et al., 2015. Background map data extracted from OpenStreetMap are available from https://www.openstreetmap.org (Openstreetmap contributors, 2023) under the Open Data Commons Open Database License (ODbL)

Replacement costs were defined based on country-based values provided by local partners for each building typology. Costs were provided in USD/m2 or, when providedy in local currency, converted following the conversion rate at the time of the calculation (Fall 2021). In order to reduce discrepancies between country-specific costs, and to provide a regionally-consistent dataset of replacement costs, we made the following assumptions:

- Given the wide range of replacement cost values collected for EMCA1, we distinguished two sub-typologies: the lower range was associated with the URM, and the upper range with RM or CM.
- For the other EMCA typologies, if a range of values was provided, we took as reference the average value.
- In the case of Turkmenistan, where costs were provided per unit of volume, we converted into cost per unit area assuming 3-meter inter-storey height
- In absence of other data, i.e., for adobe and steel typologies, we used the costs estimated by Pittore et al. (2020)

Based on these considerations, we harmonized costs making sure that the relative costs ratio between less costly construction (e.g. URM) and more expensive ones (e.g. RC frames or shear walls) are reasonable. In particular, the costs ratio between EMCA2 and EMCA1 (averaged across the two sub-typologies) does not exceed the value of 3, with the exception of Turkmenistan, where the ratio is much lower. The typology for which there are larger discrepancies across countries is the

EMCA5 (wood), likely because of the different availability and cost of the material. This is very evident in particular for Turkmenistan, where wood buildings are the most expensive.

Table 2 shows the residential building typologies and the country-based costs. For the case of EMCA1, given the strong
differences between URM and CM/RM, two sub-typologies were identified. Replacement costs are referred here to the structural cost, while the content costs were estimated as 50% of the building structural cost, following the procedure described in the HAZUS inventory technical manual (FEMA, 2021). Costs for each building unit are then found by multiplying the average reconstruction cost by the average building area for each typology, similarly to Arup (2016).

**Table 2:** Country-based replacement costs expressed in USD/m2 for each building typology and sub-typology in Central Asia.

| Typology | Sub-typology | Replacement cost (USD/m2) | | | | |
|---|---|---|---|---|---|---|
| | | Kazakhstan | Kyrgyz Republic | Tajikistan | Uzbekistan | Turkmenistan |
| EMCA1 | URM1, URM2 | 190 | 175 | 175 | 175 | 105 |
| | CM, RM-M, RM-L | 300 | 300 | 300 | 285 | 150 |
| EMCA2 | RC1,2,3,4 | 570 | 400 | 425 | 400 | 180 |
| EMCA3 | RCPC1,2 | 425 | 425 | 425 | 400 | 180 |
| EMCA5 | WOOD1,2 | 330 | 330 | 177.5 | 300 | 648 |
| EMCA4 | ADO | 125 | 125 | 125 | 190 | 125 |
| EMCA6 | STEEL | 175 | 175 | 175 | 175 | 175 |

Residential buildings exposure layers were validated for each country based on data provided by local partners for specific cities. Such data were not used in the development of the exposure layer because they only contained the total number of households, but not the building types. Differences were lower for Kyrgyz Republic and Tajikistan (which are the countries
where Pittore et al., 2020 deployed most field surveys), and larger in Uzbekistan for which the country-based building census showed largest discrepancies with the previously available information. A comparison of the fraction of building types was made for the city of Ashgabat (Turkmenistan), for which the approximate percentage of buildings of each type was available. The 65% of buildings are constituted by load-bearing masonry, while a 35% is reinforced concrete (pre-case or cast in situ). The comparison with the exposure dataset developed here shows a good agreement with differences smaller
than 5% between building fractions.

**3.3 Development of exposure layers for 2080**

Exposure layers for 2080 were developed based on three SSPs defined for Central Asia (Pedde et al., 2019). The three selected scenarios envisage socio-economic development based, respectively, on three main drivers: sustainability, unequal
investments and economic disparities and exploitation of fossil fuels together with increased energy consumption (SSP1, SSP4 and SSP5, respectively). The choice of the aforementioned SSPs was taken within the working group with the intention to  highlight the larger variations expected, and to create an upper and lower bound for expected exposure changes with respect to more 'middle of the road' scenarios (e.g. SSP2). The choice of SSP1 was also motivated by the willingness to highlight the role of governance and international cooperation, which was promoted by national-scale and international
workshops during the onset of the research project (Peresan et al., 2023). The projected exposure layers are developed starting from the population and residential buildings' exposure layers developed in this work (2.1 and 2.2). In particular, SSPs are used to inform changes in population and allocation of residential buildings.

The projected population is  estimated by decreasing/increasing the population according to the future population trends expected under each scenario. Expected population trends were extracted from the IIASA SSP database
(https://tntcat.iiasa.ac.at/SspDb/dsd?Action=html page page=about) which provide country-based indicators based on the

studies of Dellink et al. (2017), Crespo Cuaresma (2017) and Samir et al. (2019). According to these studies, the population is expected to decrease between 3 up to more than 50%, with the exception of Tajikistan where, for SSP4 scenario, population is expected to increase. In order to obtain the projected population layer, the country-based increase or decrease was applied on a cell-by-cell basis to the total population value, maintaining constant the gender and age fractions.

Despite the expected population decrease, Central Area is expected to undergo a progressive urbanization (Chen et al., 2020), associated with a strong GDP increase (see IIASA SSP database for details). This process had already started in the 2000s with an average cities growth rate of 9 to 11% (UNESCAP, 2013). Future urbanization is also assumed to be associated with a modification of building typologies, with the progressive substitution of deprecated building types in favor of modern ones. The projected residential buildings layer is thus developed by modifying the number and typology of

buildings. This process was simulated using simple rules, defined based on expert judgment provided by practitioners during 5 country-based capacity building workshops organized in Central Asia (Peresan et al., this volume). In particular, unreinforced masonry and adobe buildings will be progressively replaced with modern masonry houses (in particular, low-rise family houses). As for new multi-family apartments, they are expected to be both reinforced concrete frames or wall type buildings, but with high level of earthquake-resistant design (RC3 and RCPC2, Table 1). Wood buildings are expected

to be constructed with modern techniques (WOOD2, Table 1), while steel buildings are assumed to remain unvaried. The conversion between the number of old and new buildings was performed using conversion factors (Table 3), obtained as the ratio between the occupants per square meters in the new and the old building type. The occupants per square meter for each typology are computed based on the average building area (Table 1). Not all types are substituted with modern ones: some are left unvaried (e.g. EMCA6) or converted into a modern typology with conversion factor 1 (which means their number is

unvaried, e.g. EMCA3, EMCA5). The same replacement rules were adopted in the whole region and for all SSPs. Buildings replacement costs are maintained constant and equal to the ones in the current exposure layer. Estimating the costs in 2080 equivalent to the current ones would be associated to a large uncertainty, given the large variability of inflation rates in the region, and could lead to unrealistic values.

**Table 3:** Conversion factors between the number of old and new EMCA building types and sub-typologies (column 1 and 4, respectively), characterized by different occupation values, used to develop the 2080 residential buildings exposure layers.

| Current exposure layer | | | 2080 Exposure layer | | | Conversion factor |
|---|---|---|---|---|---|---|
| Current building type | Average occupants per building | Average occupants per square meter | 2080 building type | Average occupants per building | Average occupants per square meter | |
| EMCA1 (URM1, URM2) | 3.8 | 0.008 | EMCA1 (RM-L) | 5.2 | 0.002 | 0.25 |
| EMCA2 (RC1, RC2) | 152 | 0.014 | EMCA2 (RC3) | 152 (unvaried) | 0.014 (unvaried) | 1 |
| EMCA2 (RC4) | 190 | 0.017 | EMCA2 (RC3) | 152 | 0.014 | 0.8 |
| EMCA3 (RCPC1) | 152 | 0.03 | EMCA3 (RCPC2) | 152 (unvaried) | 0.03 (unvaried) | 1 |
| EMCA4 | 5.2 | 0.052 | EMCA1 (RM-L) | 5.2 (unvaried) | 0.002 | 0.04 |
| EMCA5 (WOOD1) | 3.8 | 0.004 | EMCA5 | 3.8 (unvaried) | 0.004 | 1 |

| | | | (WOOD2) | | (unvaried) | |
|---|---|---|---|---|---|---|
| EMCA6 | 3.8 | 0.002 | EMCA6 (unvaried) | 3.8 (unvaried) | 0.002 (unvaried) | 1 |

The exposed value of residential buildings Central Asia is therefore expected to vary due to the population variation and the progressive buildings replacement. The calculation is done in each point of the 500m regular grid. First, the number of buildings in the current exposure layer is converted into the corresponding buildings to accommodate the 2080 projected population. Then, deprecated typologies are converted into modern ones based on conversion factors. The calculation was performed making sure that the population and residential buildings values in the projected database is realistic and, in particular, that no points were associated with negative population.

The modification of the building stock is assumed to happen only in areas which are expected to be urban by 2080, while no changes are applied to the building stock in rural areas. Current urban areas were extracted from the GHSL dataset (JRC, 2021) for the latest available year (2015). The dataset comprises 7 classes that were simplified into three main ones: rural, sub-urban (which includes sub-urban and peri-urban areas) and dense urban areas. Urban areas in 2080 were identified based on the urban development trends provided by Chen et al. (2020) under the three different SSPs. Table 4 summarizes the expected urban area changes at the national scale between the GHSL dataset and the projections of Chen et al. (2020) under each considered SSP.

| Country | SSP | Urban area in 2015 (Km2) | Urban area in 2080 (Km2) | Difference (%) |
|---|---|---|---|---|
| Kazakhstan | SSP1 | 1722 | 4761 | 176 |
| | SSP4 | 1722 | 4706 | 173 |
| | SSP5 | 1722 | 4582 | 166 |
| Kyrgyz Republic | SSP1 | 359 | 687 | 91 |
| | SSP4 | 359 | 671 | 87 |
| | SSP5 | 359 | 657 | 83 |
| Tajikistan | SSP1 | 504 | 698 | 38 |
| | SSP4 | 504 | 675 | 34 |
| | SSP5 | 504 | 665 | 32 |
| Uzbekistan | SSP1 | 3279 | 4529 | 38 |
| | SSP4 | 3279 | 4379 | 34 |
| | SSP5 | 3279 | 4365 | 33 |
| Turkmenistan | SSP1 | 419 | 776 | 85 |
| | SSP4 | 419 | 736 | 76 |
| | SSP5 | 419 | 697 | 66 |
| TOTAL | SSP1 | 6283 | 11451 | 82 |
| | SSP4 | 6283 | 11167 | 78 |
| | SSP5 | 6283 | 10966 | 75 |

**Table 4.** Urban area in 2015 (derived from the GHSL database) and 2080 (derived from the Chen et al., 2020 database) and percentage difference in each country and under each SSP.

Table 4 shows that a strong urbanization is expected in all Central Asia countries between 2020 and 2080. The largest variations are expected in Kazakhstan, where the urban area is expected to increase of more than 160% under the three SSPs. Substantial changes are also expected in Kyrgyz Republic (between 80 and 90%) and Turkmenistan (between 65 and 85%). Lower percentages are found in Tajikistan and Uzbekistan, ranging between 30 and 40%. Comparing sub-urban and urban areas in 2015 with the ones for 2080, we identified areas which, under the three different SSPs, are expected to be urban in 2080. This includes areas that were already classified as urban in 2015, but also areas that are expected to become so between 2015 and 2080, and where the building stock will undergo progressive replacement. In urban areas, abandoned or unoccupied structures are assumed to be demolished during the buildings replacement process. In rural areas the deprecated building types are maintained in the 2080 exposure layer in order to avoid underestimating the risk related to weak typologies which might still be in use, or not demolished, despite their age.

## 4. Results

### 4.1 Population exposure

Figure 2 shows the population layer produced at 100m resolution at regional scale for an urbanized area in Aqtöbe (Kazakhstan). In each point of the grid, the total population and the number of men, women, elder and young population (over 60 and under 5 years old, respectively) are provided. Total values were also computed for each country and Oblast.

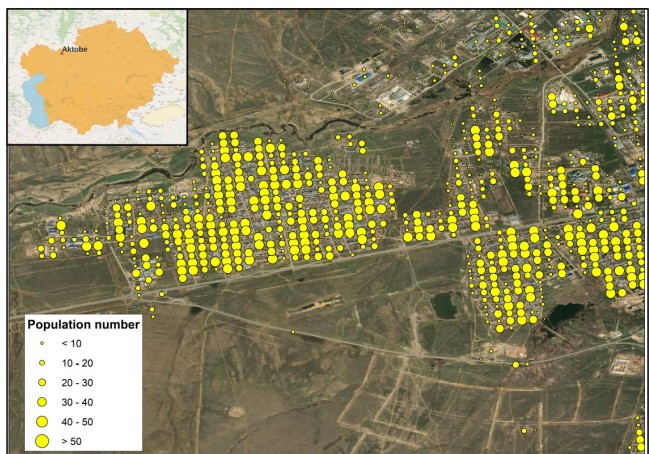

**Figure 2: Example of the population layer produced at 100m resolution for a selected area in the town of Aqtöbe in Kazakhstan.** The figure shows an urbanized area with different uses (industrial, in the top of the image, residential, in the center of the image, and rural with low or null population density). Background map data extracted from OpenStreetMap are available from https://www.openstreetmap.org (Openstreetmap contributors, 2023) under the Open Data Commons Open Database License (ODbL)

### 4.2 Residential buildings exposure

Fig. 3 shows the spatial distribution of a) one sub-typology of the EMCA1 typology (Table 1), the unreinforced masonry (URM), b) one sub-typology of the EMCA2 typology (Table 1), the low-rise reinforced concrete buildings constructed before 2006 (RC1) and c) one sub-typology of the EMCA3 typology (Table 1), the precast reinforced concrete buildings constructed before 1980 (RCPC1). Map shows the spatial distribution of buildings in the entire Central Asia region and for one selected study area, at 500-m resolution. Similar maps can be produced for other building typologies.

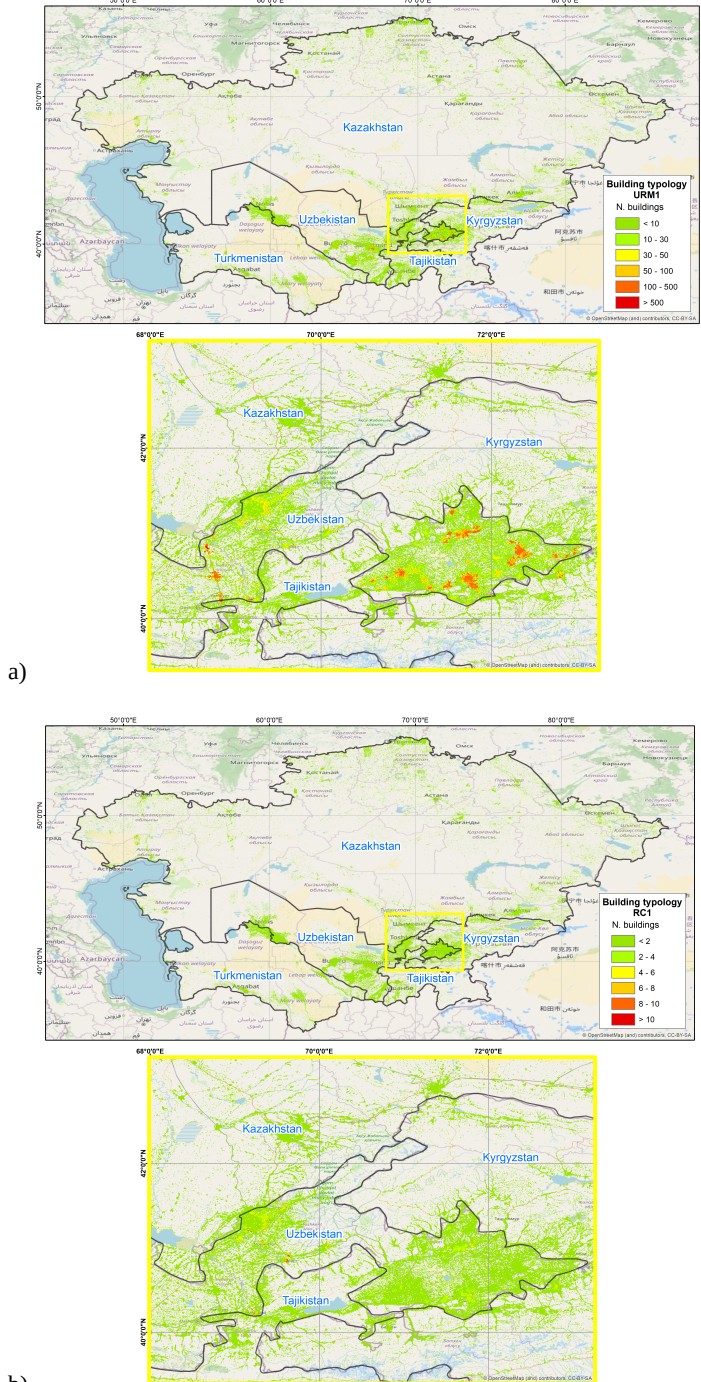

a)

b)

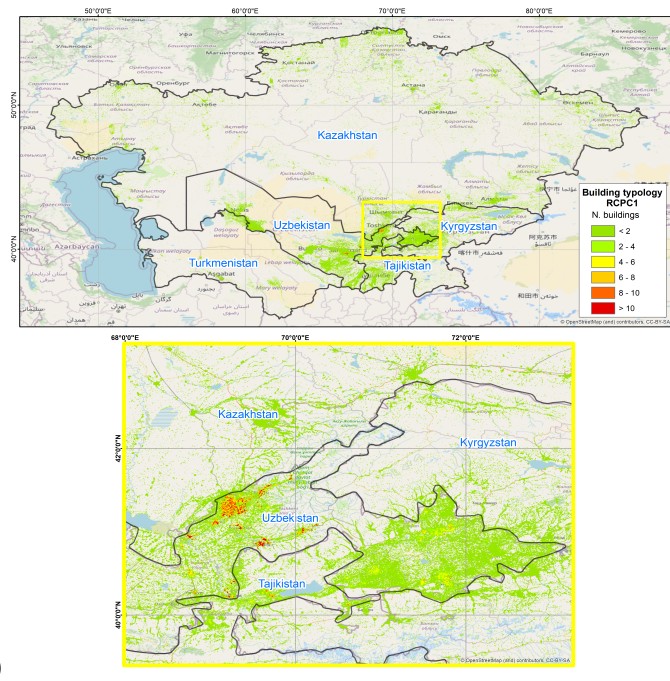

c)

**Figure 3.** Number of buildings in each 500-m cell belonging to a) the URM1 sub-typology of EMCA1, b) the RC1 sub-typology of EMCA2 and c) the RCPC1 sub-typology of EMCA3. Each maps is displayed for the entire Central Asian region (top) and for a selected area (bottom). Map data from OpenStreetMap available from https://www.openstreetmap.org (Openstreetmap contributors, 2023) under the Open Data Commons Open Database License (ODbL)

Table 5 provides the total number of exposed buildings per typology and country and their associated structural cost expressed in Billion USD. The total structural cost of residential buildings in Central Asia is of approximately 1,200 Billion USD, and the higher fraction is associated with Uzbekistan and Kazakhstan (the 62 and 29%, respectively).

**Table 5.** Total number of residential buildings in each EMCA typology and the total structural cost for each country and for Central Asia (in billion USD).

| Country | Residential buildings | EMCA1 | EMCA2 | EMCA3 | EMCA4 | EMCA5 | EMCA6 | Structural cost (Billion USD) |
|---|---|---|---|---|---|---|---|---|
| Kazakhstan | 2,378,980 | 614,196 | 41,031 | 35,243 | 821,613 | 669,169 | 197,693 | 356 |
| Kyrgyz Republic | 592,637 | 196,419 | 2,647 | 4,216 | 384,169 | 4,702 | 467 | 35 |
| Tajikistan | 844,336 | 218,439 | 2,226 | 10,939 | 607,539 | 4,582 | 599 | 58 |
| Uzbekistan | 5,708,009 | 4,790,954 | 64,795 | 122,579 | 567,415 | 145,899 | 16,330 | 773 |
| Turkmenistan | 280,358 | 97,760 | 10,357 | 6,989 | 158,785 | 5,887 | 567 | 20 |
| Central Asia | 9,804,432 | 5,917,768 | 121,056 | 179,966 | 2,539,521 | 830,239 | 215,656 | 1,242 |

Figure 4 shows the structural replacement cost fraction of building typologies in the 5 considered countries. The greatest contribution to the total costs comes from EMCA1 (Masonry) followed by EMCA3 (Precast reinforced concrete) in all
countries.

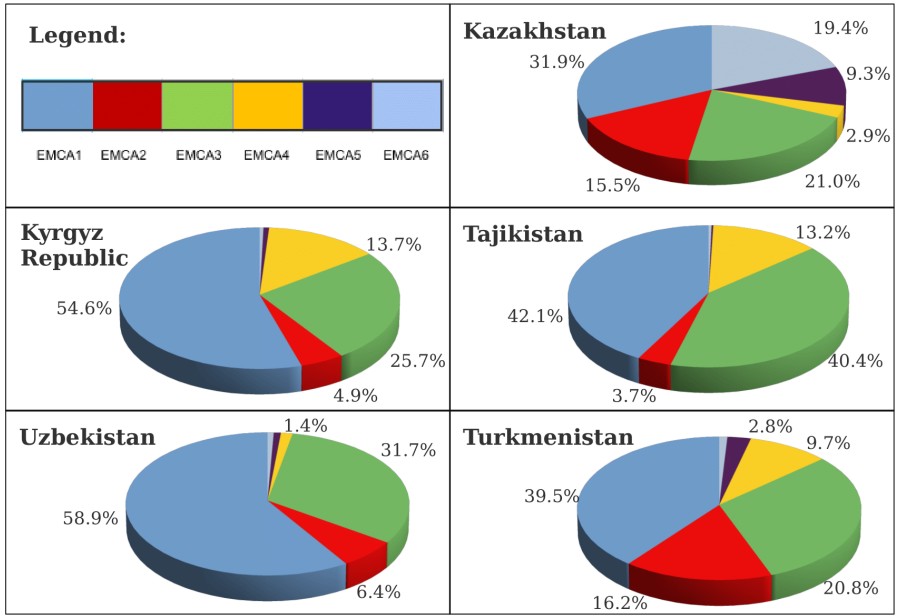

**Figure 4**: Fraction of replacement costs (expressed in percentage of the total replacement costs) associated with each building typology (EMCA 1 to 6) in the 5 Central Asia countries.

### 4.3 Exposure layers for 2080

Table 6 provides the total projected buildings number and the associated replacement costs for 2080 and the % variation (total and per capita) with respect to the layer developed for the present time (2021). Figure 5a shows the percentage cost variation with respect to the current total replacement costs for each considered scenario. Costs are expected to decrease for Kazakhstan and Uzbekistan and increase in Kyrgyz Republic and Turkmenistan. Kyrgyz Republic and Tajikistan show both increase and decrease, depending on the considered scenario. The average replacement cost per capita in each country is nonetheless
expected to increase for most countries and scenarios due to the population decrease and the adoption of building types associated with a higher replacement cost. The higher residential buildings replacement cost per capita is expected in Tajikistan under the SSP5 scenario (which is associated with the stronger population decrease). Note that the overall number of buildings is expected to decrease under all scenarios, due to the fact that unoccupied buildings are not maintained in urban areas, where they are replaced with other building typologies.

The replacement cost associated with some typologies, in particular EMCA1 and EMCA2, is expected to increase (Fig. 5b). This is due to the progressive replacement of buildings belonging to deprecated typologies with more recent ones. In particular, URM buildings are replaced with RM which has higher costs per square meter. Similarly, EMCA2 buildings of type RC4 are replaced with RC3 with a conversion factor of 0.8, for which there is a larger number of buildings and subsequently a higher total replacement cost. The expected increase is larger in Kyrgyz Republic and Tajikistan. The only
typology associated with a negative cost variation is EMCA4, because part of the buildings are replaced with EMCA1 typology. EMCA types 5 and 6 are not included as they are not expeected to undergo changes.

**Table 6.** Total residential buildings and expected percentage variation (columns 3 and 4, respectively) and total replacement costs estimated for 2080 and expected replacement cost % variation between 2080 and current exposure layer, total and per capita (column 5, 6 and 7, respectively). Values are shown for the three considered SSPs.

| Country | Scenario | Total buildings | Building number % variation | Replacement costs in 2080 (Billion USD) | Replacement costs (% variation) | Replacement costs per capita (% variation) |
|---------|----------|-----------------|-----------------------------|------------------------------------------|----------------------------------|---------------------------------------------|
| Kazakhstan | SSP1 | 2,110,243 | -11.3 | 330 | -5.7 | -2.8 |
| | SSP4 | 2,111,560 | -11.2 | 336.4 | -3.9 | 5.6 |
| | SSP5 | 2,116,377 | -11.0 | 306.7 | -12.4 | -10.6 |
| Kyrgyz Republic | SSP1 | 524,066 | -11.6 | 29 | -6.5 | 11.4 |
| | SSP4 | 525,312 | -11.4 | 32.3 | 4.2 | 28.6 |
| | SSP5 | 525,805 | -11.3 | 32.3 | 4.2 | 51.0 |
| Tajikistan | SSP1 | 790,097 | -6.4 | 54 | -3.6 | 25.2 |
| | SSP4 | 799,681 | -5.3 | 56.2 | 0.4 | -20.4 |
| | SSP5 | 784,708 | -7.1 | 55.5 | -0.9 | 125.2 |
| Uzbekistan | SSP1 | 4,230,863 | -25.9 | 681 | -11.1 | 3.4 |
| | SSP4 | 4,256,952 | -25.4 | 688.3 | -10.1 | 9.6 |
| | SSP5 | 4,264,616 | -25.3 | 688.7 | -10.1 | 19.9 |
| Turkmenistan | SSP1 | 266,390 | -5.0 | 19 | 0 | 12.4 |
| | SSP4 | 266,305 | -5.0 | 19.9 | 4.7 | 24.7 |
| | SSP5 | 267,064 | -4.7 | 19.9 | 4.7 | 26.2 |
| Central Asia | SSP1 | 7,921,659 | -19.2 | 1113 | -8.9 | 3.8 |
| | SSP4 | 7,959,810 | -18.8 | 1133.1 | -7.3 | 3.3 |
| | SSP5 | 7,958,570 | -18.8 | 1103.1 | -9.7 | 17.1 |

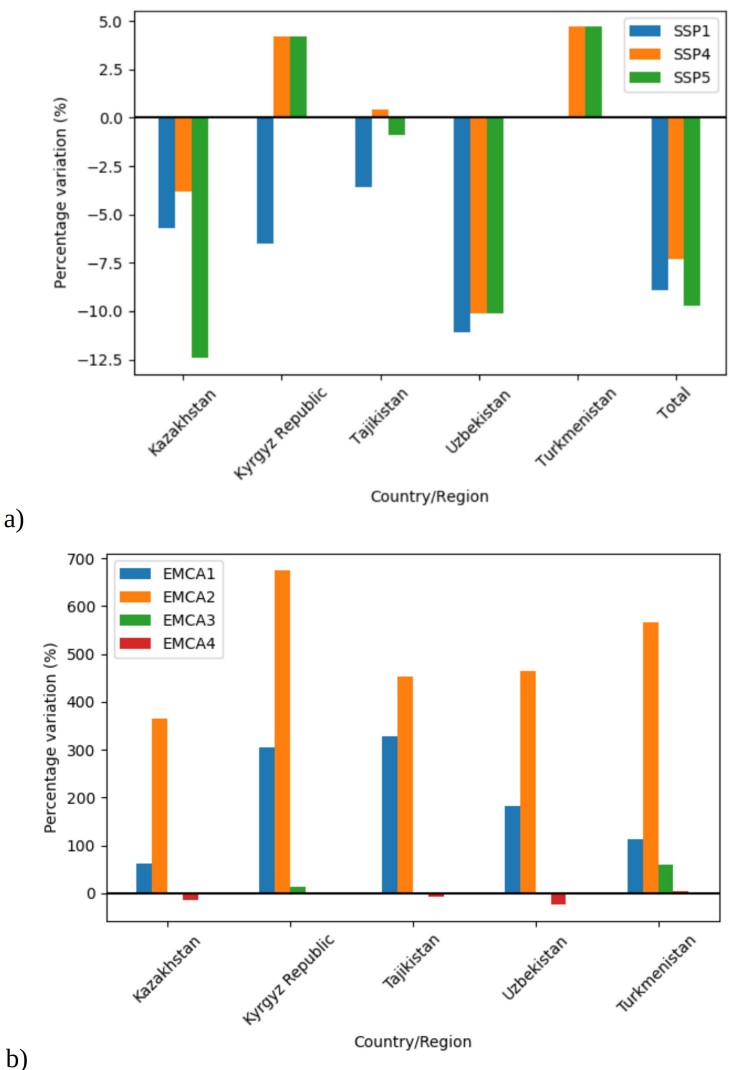

a)

b)

**Figure 5:** Buildings replacement cost variation (expressed in percentage variation with respect to the replacement cost in the current exposure layer) estimated a) for the total replacement costs in each considered scenario (SSP1, SSP4 and SSP5) and b) for the total replacement cost of each EMCA typology averaged across the three SSPs.

## 5. Discussion

The regionally-consistent exposure database presented here is based on the combination of global and regional layers and data collected at national and sub-national scale. Performing a regionally-consistent exposure development requires the harmonization of the exposed assets characteristics, which might vary within the study area. There are nonetheless several challenges associated with the definition of a regionally-consistent exposure database, in particular related to the scattered and inhomogeneous information available (e.g. different time coverage or spatial resolution for the 5 countries in the region). In particular, the spatial disaggregation process, if based on datasets from different years, can introduce inconsistencies which can be identified by consistency checks among the variables which have known relation (e.g. building occupancy and

population). In this work we tried to achieve an optimum balance between the different data availability and reliability, in order to grasp the differences and peculiarities of the 5 countries. Population layers were developed at 100m resolution while residential buildings were aggregated at 500m resolution. Both layers can be resampled for the purpose of more specific analyses, but this should be done carefully and integrating specific information which might become available in the future.

Final data, together with metadata and description, is provided in the GED4All format (Global Exposure Database for Multi-Hazard Risk Analysis, Silva et al., 2022) developed by the Global Facility for Disaster Reduction and Recovery (GFDRR) in order to supports risk analyses.

With regards to population and buildings data, further efforts should be devoted to identify procedures to automatically collect and update census data, reducing the effort of in presence surveying. In addition, our analysis does not account for
many aspects, such as night-day occupation patterns and socio-economic exposure, which can be included in future in the analysis (e.g. Freire and Aubrecht, 2012). Note that this information is already envisaged in the GED4All taxonomy but scarcely available for many areas at risk worldwide.

As for residential buildings, common but broad typologies were defined, also based on previous projects, so that they are valid across the entire region. Such typologies are associated with sub-typologies that can be analyzed further in the future.
In addition, emerging building typologies should be also included (e.g. new type of constructions based on lightweight insulated panels). The fraction of building typologies within the building stock was extracted from national census, when available. However, some census only provided the number of households, and required converting them into buildings, assuming an equivalent number of households per building type. In addition, building typologies are defined by different classes in country-based census (e.g., some distinguish between material of walls and of the load-bearing structure, other
don't). The process of combining different census can nonetheless lead to discrepancies. Finally, sub-typologies can be quite different in the Central Asia countries due to multiple factors, including different constructive tradition, climatic zone and other cultural aspects, that should be taken into account in future work. For this reason, a common protocol of data collection could be extremely beneficial both for single countries and for regional-scale approaches.

Despite the overall generalization required to develop a regional-scale exposure database, relevant differences were
maintained using country-based buildings replacement costs. A comparison with costs provided by Arup (2016) for Kyrgyz Republic shows that costs of EMCA1, 2 and 3 are quite similar but replacement costs of other typologies such as timber and steel have varied, which is probably due to the variations suffered by the raw material price. This is the first attempt to collect construction costs for each of the 5 countries of Central Asia, but any financial assessment should be carried out based on detailed and updated information. The difficulty of gathering unit costs (e.g. material costs, labor costs) and costs
associated with the reconstruction process (e.g. debris removal) was also subject of discussion during the workshops organized with local stakeholders (Peresan et al., 2023). Future work in the region can benefit from a similar process, with specific workshops involving academics, practitioners and other stakeholders to assess replacement and reconstruction costs. As for content costs, they were estimated following the HAZUS methodology (FEMA, 2021) as a function of the structural cost of each building typology. HAZUS is widely adopted and, in absence of region- or country-specific data on
content costs, was assumed to be applicable to Central Asia.

In this work, we provided a projection for 2080 based on the combination of three SSPs defined for Central Asia (Pedde et al., 2019). The choice of the SSPs was taken by the working group and subjected to both subjective and practical considerations (e.g. the number of risk scenarios to be performed based on that), but should be integrated in the future with other SSPs. The development of such projections is based on a number of assumptions, the main ones being that the
population decrease is assumed to happen homogeneously in each country, that the renovation of building stock follows the same rules in all urban areas of Central Asia and that replacement costs do not vary. Also, that deprecated buildings remain in the building stock only in rural areas while are replaced in urban areas, which partially justifies why the overall number of buildings is decreasing, However, this might lead to an overall underestimation of exposure and subsequent risk because abandoned buildings can suffer damages and cause direct and indirect losses to the society (e.g. by collapsing and blocking a

road). This results in a simple projection that does not account for the complex dynamics behind socio-economic development. In  fact, our procedure the choice of SSPs only influences few indicators related to exposure (e.g, population and number of buildings). The uncertainties related to the projection of economic indicators to 2080 should be taken into account when using such projections for assessing risk. These projections are in fact intrinsically associated with a large uncertainty widely discussed in the academic literature (e.g., Dellink et al., 2017). Also, according to Dellink et al. (2017),

despite the GDP is overall expected to grow at global scale, the GDP growth rates and income growth rates are expected to lower sometime between 2030 and 2040 for all SSPs. Thus, the GDP growth values do not necessarily provide a realistic economic growth indicator for the region. In addition, different SSPs would favor different economic systems (e.g. based on fossils fuels rather than on sustainable technologies), which in their turn would influence the type and number of buildings and their location. The proposed projections should therefore be improved in the future by including national and regional

strategies and development plans and by updating exposure layers accordingly. More sophisticated analyses should account also for the different economic system envisaged in each SSP and the socio-economic consequences of its adoption, and potentially assess the deeper implications of different SSPs for exposure assessment. Finally, the SSPs presented here rely on indicators such as population change, urbanization rate and GDP, which are not independent. Future work should explore the interplay between the population change and the urbanization process, and how they affect exposure and risk indicators. The

projections might as well be complemented by urban simulation modeling for selected cities or Oblasts.

The regional-scale dataset of population and residential buildings provided here can support further analyses on the expected damages and risks caused by hazardous phenomena such as earthquakes and floods. However, the building typologies included in this exposure model were originally define for earthquakes and do not account for all the characteristics deemed relevant for flood vulnerability. The use of this exposure model for the assessment of flood-induced risks should be therefore

done carefully, especially when using it at sub-national scale. A classical multi-hazard approach (i.e. using different vulnerability functions for each building class in the exposure model, such as in Coccia et al., 2023) could be complemented with other approaches that account for cumulative damage such as, for example, earthquake and tsunami (Gomez Zapata et al., 2022). Residential buildings are very relevant for disaster risk reduction as they host a large fraction of population, in particular during night time, and are responsible of a large fraction of life losses during earthquakes. Also from the financial

point of view, a comparison between the exposed value of residential building with respect to other  types (commercial, industrial, healthcare and educational) shows that residential buildings account for the largest fraction (between 47% and 76%) of replacement cost of all building types. The 2080 layers presented here offer a starting point for  the definition of risk mitigation strategies. For example, they can help identifying the typologies that are more prone to generate losses and/or to generate financial risk. Under these considerations, they might be replaced in the future  with less vulnerable residential

building typologies, as envisaged by many expert and practitioners in the region during exposure development workshops (Peresan et al., 2023).

The work presented here relies on assumptions that are needed in order to produce results at the regional scale. In particular, country-based data are paramount in order to enhance the regional-scale datasets with the specific characteristics of exposed assets. In addition to official data sources, experts opinion was collected on a number of aspects for which  data were not

available or incomplete. For example, they provided information on census building typologies and their correspondence with EMCA typologies and on their construction costs. They also informed on which building typologies are being gradually replaced in the building stock, supporting the development of future exposure layers. This was made possible by the organization of 5 country-based exposure workshops (Peresan et al., 2023) which enhanced the interaction with local experts, practitioners and representatives of governments. Interactions with local experts are indispensable in order to

identify, gather and interpret correctly the different data sources that concur to the development of reliable exposure layers.

**6. Conclusions**

This work produces the first high-resolution regionally-consistent exposure database of population and residential buildings exposed to floods, earthquakes and landslides in Central Asia. The dataset comprises exposure layers for 2020 and 2080, developed under different SSPs. Results of the exposure assessment show that the residential buildings in central Asia are distributed heterogeneously, with large differences between urban and rural areas. We also assessed the value of exposed buildings in Central Asia in terms of replacement costs, of which a large fraction is located in Uzbekistan and Kazakhstan which are the larger and more populated countries. The 2080 exposure projections show that, despite a general population decrease, a strong urbanization and economic growth is expected in Central Asia, with subsequent increase of the replacement cost per capita. The regional-scale exposure database produced during this project can act as a starting point for current and future disaster risk mitigation activities devoted to reducing physical, socio-economic and financial impacts of natural hazards in Central Asia.

**Data Availability**

Facebook high resolution population data for each Central Asia country is available at https://data.humdata.org/. The regional-scale layer of Pittore et al. (2020) is available at https://github.com/GFZ-Centre-for-Early-Warning/EMCA-Exposure. The Global Human Settlement Layers are available at https://ghsl.jrc.ec.europa.eu/ at 1km resolution for the years 2000 and 2015. Spatial layers of expected urban area in 2080 under different SSPs (Gao and O'Neill, 2020) and are available at https://dataverse.harvard.edu/dataverse/geospatial_human_dimensions_data. The spatial layers of exposure for population, residential buildings and 2080 projections developed in this work are available at the World Bank data portal (https://datacatalog.worldbank.org/search/dataset/0064117/Central-Asia-Exposure-Data) together with the technical reports produced during the SFRARR project under the Creative Commons Attributions 4.0 license. Data are associated with metadata following the Ged4ALL system (http://riskdatalibrary.org/resources).

**Acknowledgments**

The project Strengthening Financial Resilience and Accelerating Risk Reduction in Central Asia was funded by the European Union and implemented by World Bank. We sincerely thank all the project team members, in particular Dr. Sergey Tyagunov, Dr. Paola Ceresa, Dr. Gabriele Coccia, Prof. Stefano Parolai and Dr. Denis Sandron, and the World Bank specialists, in particular Dr. Stuart Alexander Fraser and Dr. Madina Nizamitdin, for their constructive contribution to the project.

**Author contribution**

CS, AT, EF developed the exposure assessment methodology, and CS and AT carried out the analyses. All co-authors contributed to the data collection and to the discussion of results. CS prepared the manuscript with contributions from all co-authors.

**Competing interests**

The authors declare that they have no conflict of interest.

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
