# Peer review of "A new regionally consistent exposure database for Central Asia: population and residential buildings"

_Natural Hazards and Earth System Sciences, 2023_

## Referee Comment (RC1)

**Manuscript title: A new regionally consistent exposure database for Central Asia: population and residential buildings**

This manuscript proposes a regional exposure database featuring the population and residential building stock for the countries of Central Asia. This new dataset has a regional scale, with the appropriate resolution to be used in multi-hazard and risk assessment. Moreover, the authors provide estimates for exposure for the year 2080 that can support current and future risk mitigation strategies. To achieve this, the authors use high resolution population and building datasets to generate gridded exposure datasets.

The introduction clearly states the research goals and main novelties. The methodological aspects of the research are very interesting. The authors rely on local data and expert knowledge to provide reliable building characterization and replacement costs. The manuscript presents relevant information for other risk scientists. This information is clearly laid out in maps, figures, and tables, making the paper useful, appealing, and easy to read. The exposure layers for 2080 are a significant addition to the research outputs. These rely on future scenarios of population and urbanization to propose different possibilities for future exposure in Central Asia. The results section is brief, clear, and very well written. I praise the authors for presenting the results of their work so well.

The manuscript contains a couple of typos, which can be easily corrected, but it is also missing several references, some of which are essential to the research. I strongly suggest thorough proof-read by the authors. Regarding the methodological aspects of the research, there are only a couple of points that need to be explicit in the manuscript. One is regarding the SSPs, specifically the motivation for the scenarios chosen and the uncertainties that they account for in this research (i.e., it seems that future population and urbanization are being accounted for, but not future sustainability, resilience, or vulnerability of the residential building stock). The other point that needs to be clearly addressed is the expected reduction in the future number of buildings, which is most likely due to the fact the future abandoned or unoccupied buildings are not being accounted for in the 2080 exposure dataset.

Based on this assessment, I believe the manuscript would be an excellent contribution to the scientific community, and that its main contents are up to the standards of NHESS. I also want to extend my congratulations to the authors for their hard work. My recommendation is to **accept the manuscript, subject to minor revisions.** Please find my comments line to line below, which I hope can help the authors greatly improve the quality of the final publication.

**Introduction**

Lines 30-35: Reference Yu et al. 2019 is missing in the references section. The authors need to include it.

Lines 45-50: Reference UNECE 2017 is missing in the references section. The authors need to include it.

Lines 55-60: The authors mention that the only regional-scale exposure dataset of residential buildings available at the time (April 2023) is provided by Pittore et al., (2020). I believe the Global Earthquake Model Foundation has provided an updated exposure model for Central Asia in terms of population, building counts and replacement costs (See Yepes-Estrada et al. 2023 and https://github.com/gem/global_exposure_model). The authors should consider mentioning the update as well.

Lines 75-80: The authors state that the first regional-scale exposure dataset for Central Asia was developed by Pittore et al., (2020) using ground-based and remote sensing datasets. The Global Earthquake Model Foundation also has updated exposure datasets covering Central Asia using a bottom-up approach (Yepes-Estrada et al. 2023). Herein, however, the authors maintain that they are proposing the 'first regionally-consistent exposure database for Central Asia'. Were the first efforts and methodologies to map the residential exposure in the region inconsistent? How so? There are novelties in the new dataset the authors propose, but this assertion needs further explanation to be included in the manuscript.

**Methodology**

Lines 120-125: The authors mention that comparing Facebook data versus national census data results in differences in the total population that exceed 20% in 7 oblasts. What is the total number of oblasts under study? What is the difference between the total population estimates from the census and Facebook? Would it be possible to elaborate? The use of Facebook data is an interesting approach and I think it would be interesting for other researchers to see how it compares to national databases.

Lines 135-140: The Global Earthquake Model building taxonomy by Brzev et al., 2013 has been updated by Silva et al., 2022 (https://doi.org/10.1007/s13753-022-00400-x.).

Lines 160-165: The authors used the Facebook population data and adjusted it to respect local estimates of the population. Then the buildings were taken from Pittore et al. (2020). Both datasets end up distributed in variable-resolution grids. These disaggregation techniques can result in an inconsistent number of occupants per dwelling and occupants per building in each cell, especially if datasets developed independently are being combined. Do the time stamps of the population and building datasets agree with each other? (i.e. the population of 2020 and the number of buildings by 2020). Was the final number of occupants per dwelling and building consistent with the subnational local estimates after the distribution? It would be a great addition to the methodology section to briefly mention the checks performed after the disaggregation methods.

**Development of exposure layers for 2080**

Lines 210-2020: HAZUS is mentioned throughout the manuscript, but I could not find the reference in the reference section (HAZUS 2021? FEMA 2021? Inventory technical manual?). This is the third reference that is either missing from the manuscript or difficult to find in the corresponding section. There is also a typo: 'manual (2021).g dife Costs'. I strongly recommend the authors to proofread the manuscript carefully.

Lines 235-245: The SSPs chosen to develop these layers were SSP1, SSP4 and SSP5. This is a very interesting choice. However, the authors mention that these scenarios envisage different development drivers, but provide no further motivation for selecting these. For example, why was the SSP2 not included? The 'middle of the road' considers less strong deviations from the current fertility trajectories, hence it is the likely scenario in terms of the future population and urbanization. Could the authors elaborate on what motivated this choice? Was supporting risk management strategies a part of this motivation? How so?

Lines 245-265: It is unclear how the SSPs are used beyond the scenarios of future population and urbanization. The uncertainty regarding the future population is taken from the SSPs. The different future population scenarios are used to infer a future number of buildings, and the urbanization layers inform location and density. However, there seems to be no uncertainty considered in the future characterization of the buildings, which influences the final vulnerability and sustainability of the future exposure datasets.

This would be a very difficult task, and the authors rightly rely on expert judgement alone to propose a single set of rules for future building characterization. There is no mention of this set of rules changing depending on the SSPs. If this is the case, the authors need to be explicitly clear in the manuscript that the SSPs only inform the population figures and building allocation. This is important given the choice of SSPs by the authors. For example, the SSP1 (sustainability with low emissions) and the SSP4 (high inequality with high emissions) are clearly opposite scenarios. For example, a reader could understand that sustainability and a more resilient built environment were at the core of creating the future building characterization for an SSP1 scenario, but it doesn't seem to be the case. I think this section really needs more clarity on this point.

**Results**

The results section is brief, clear, and very well written. I praise the authors for presenting the results of their work so well. I would even recommend the authors include a figure like Figure 3 for all the sub typologies but only if it is possible within their time constraints.

The only comment here is regarding Table 5. The building variation is mostly negative in the scenarios, given the population prospects of Central Asia. However, it was not explained in the methodology why the building count is decreasing. I believe that it is because the dataset is meant to include occupied structures only and abandoned or unoccupied structures are not being considered. This needs to be explicit in the methodology.

This is important because a sustained population decrease does not always lead to a sustained decrease in the exposed number of structures. For example, despite a steady decline in population, the number of total dwellings exposed in Albania rose by more than 25% between 2001 and 2011 ([www.instat.gov.al](www.instat.gov.al)). In other words, residential exposure in Albania (new dwellings become new buildings) is increasing despite the decreasing population, as there are other factors beyond population driving changes in residential exposure. Moreover, less population might lead to fewer occupied structures, but abandoned or unoccupied structures are still at risk of flooding, shaking, and causing debris, which will incur an economic loss for society. If the datasets the authors proposed account only for occupied structures, that is perfectly reasonable, but the limitations of this approach should be mentioned as well.

**Discussion**

Lines 345-350: The authors mention that the final data is provided in GED4ALL format. Is it open access already? Consider using 'will be provided'.

**References**

Please include the missing references

---

## Author Comment (AC1)

ANSWER TO REVIEWERS

REVIEWER 1

The paper presents a new exposure model providing information on population and residential buildings in five Central Asian Countries, namely Kazakhstan, the Kyrgyz Republic, Tajikistan, Turkmenistan and Uzbekistan. The model, mainly aiming at supporting seismic risk assessment, is updating and further advancing a former model developed within the framework of the EMCA (Earthquake Model Central Asia) in several aspects. At regional and transnational scale the authors improved the spatial resolution of the model by leveraging high-resolution proxies obtained by open-source data. At national scale up-to-date authoritative have been also considered and integrated by expert consultations. This is already a significant and useful achievement considering the challenges related with collecting and integrating information on exposure in data-scarce regions. Furthermore, the authors projected the updated model to a relatively far future (2080) to investigate the possible changes in exposure in the region based on different SSPs and related urbanization models. Although such projection is likely affected by strong uncertainties, it would still be very useful for risk assessment under non-stationary conditions. In fact, although seismic hazard can be considered stationary over the considered time-frame, the dynamics of exposure already proved to be instrumental in driving the expected risk over the next decades and a better consideration of such dynamics might improve both short- and longer-term risk mitigation and climate change adaptation efforts. The authors in particular estimate the expected relative differences in building replacement cost (considering no variation in usd/m2) between current and future, which provides useful insights on the possible change in seismic risk, but fall a bit short in exploring the interplay between the population change and the urbanization process, which would perhaps allow further considerations on the possible spatial pattern of future risk in the region. Overall the paper is well designed and written and would provide a interesting and useful contribution to the topic of exposure modelling.

*Thank you very much for your suggestions. We replied to all the points raised by the reviewer. We emphasized the importance of tackling exposure variability to improve risk assessment methodologies and acknowledged the limitations of the approach presented here. Please find attached our responses to each comment.*

Detailed comments / clarifications needed

line 161 - is the population value used only to spatially distribute buildings and the number of buildings is provided at the oblast scale by the most recent housing census?
*Yes, recent building census data was collected for Kazakhstan and Uzbekistan and the number of buildings in each typology was distributed on the original model of Pittore et al (2020). The population value was used to spatially distribute buildings. We specified this in the manuscript.*

line 220 - typo ("g dife")
*The typo was corrected*

note: in general I find the term "replacement cost" better than "reconstruction cost" since in practice in case of reconstruction (after an event) the building typologies, building pratices and used materials would be the same.
*We understand your comment and we agree with you. The term 'reconstruction cost' was used here after the suggestion of the World Bank analysts because they wanted to give emphasis to the cost associated with the reconstruction process and to the potential of reconstructing buildings using more recent technologies and avoiding deprecated typologies. However, we did not explicitly account for unit costs and/or assess additional expenses such as debris removal, fees or material cost in the cost per square meter provided by local partners. We therefore renamed to 'replacement cost' through the manuscript. The difficulty of gathering such data and of defining the reconstruction cost was also subject of discussion during the workshops organized with local stakeholders. We included a sentence on the limitations associated with assessing the reconstruction costs in the discussion, and underlined the importance of gathering such costs with specific procedures, for example workshops involving practitioners and stakeholders.*

Lines 276 - 284: it would useful to provide a more precise idea of the expected change in urbanization based on the Chen et al. model for the three considered SSPs, including (in the results section) for instance a table with the difference in the estimated values of the different urbanized areas in the different countries and according to the various SSPs.

*We included a table that summarizes the expected urban area changes at the national scale between the GHSL dataset and the projections of Chen et al. (2020). The largest variations are expected in Kazakhstan, where the urban area is expected to increase of more than 160% under the three SSPs. Substantial changes are also expected in Kyrgyz Republic (between 80 and 90%) and Turkmenistan (between 65 and 85%). Lower percentages are found in Tajikistan and Uzbekistan, ranging between 30 and 40%. The table is found below and was included in the manuscript:*

| Country | SSP | Urban area in 2015 (Km2) | Urban area in 2080 (Km2) | Difference (%) |
|---|---|---|---|---|
| Kazakhstan | SSP1 | 1722 | 4761 | 176 |
| | SSP4 | 1722 | 4706 | 173 |
| | SSP5 | 1722 | 4582 | 166 |
| Kyrgyz Republic | SSP1 | 359 | 687 | 91 |
| | SSP4 | 359 | 671 | 87 |
| | SSP5 | 359 | 657 | 83 |
| Tajikistan | SSP1 | 504 | 698 | 38 |
| | SSP4 | 504 | 675 | 34 |
| | SSP5 | 504 | 665 | 32 |
| Uzbekistan | SSP1 | 3279 | 4529 | 38 |
| | SSP4 | 3279 | 4379 | 34 |
| | SSP5 | 3279 | 4365 | 33 |
| Turkmenistan | SSP1 | 419 | 776 | 85 |
| | SSP4 | 419 | 736 | 76 |
| | SSP5 | 419 | 697 | 66 |
| TOTAL | SSP1 | 6283 | 11451 | 82 |
| | SSP4 | 6283 | 11167 | 78 |
| | SSP5 | 6283 | 10966 | 75 |

Table 5 / Fig 5: it would interesting to provide or discuss variations in the replacement/reconstruction costs for the different building typologies (which in turn depend on the interplay between population and changed urbanization).

*Yes, the variation of reconstruction costs depends on the urbanization, because the replacement of deprecated typologies happens only in areas that are expected to be classified as urban in 2080. We estimated the percentage variation of reconstruction costs for each EMCA typology. The variation was computed for each SSP and also on the average values obtained for the three SSPs, as follows.*

*Percentage variation for each EMCA typology, SSP and country:*

[Figure]

*Percentage variation for each EMCA typology and country averaged over the three SSPs:*

[Figure]

*The figure showing the  average variations was added as Fig. 5b in the revised manuscript.*

*The larger variation is expected for EMCA2, followed by EMCA1. Both typologies are expected to undergo a progressive replacement with more recent typologies. In particular, URM buildings are replaced with RM which has higher costs per square meter. Similarly, EMCA2 buildings of type RC4 are replaced with RC3 with a conversion factor of 0.8, for which there is a larger number of buildings and subsequently a higher total replacement cost. The variation is larger in Kyrgyz Republic and Tajikistan where in particular the cost associated with the EMCA1 type sees a stronger increase. The larger differences in reconstruction costs between SSPs are seen for Tajikistan, as also mentioned when commenting the general results. The only typology associated with a negative cost variation is EMCA4, because part of the buildings are replaced with EMCA1 typology. EMCA types 5 and 6 are not included as they not suffer changes. These considerations were added to the manuscript together with the new figure.*

Line 377 - typo ("us")
*The typo was corrected*

Line 389 - it is indicated a possible application in case of floods, but the building typologies are specifically targeted at earthquake risk. The authors should warn that, although the model can be considered for other hazards, it might be sub-optimal or anyway an euristic or probabilistic mapping should be considered to fit to other types of vulnerabilities (see, e.g., Zapata et al., https://nhess.copernicus.org/preprints/nhess-2022-183/nhess-2022-183.pdf)
*Yes, we agree with your comment. In principle, the exposure model should support regional-scale risk assessment, but it would be simplistic to apply this model at sub-national and local scale because, despite its resolution is higher, it does not account for all the characteristics deemed relevant for assessing impacts caused by floods. A classical multi-hazard approach (i.e. using different vulnerability functions for each building class in the exposure model, such as in Coccia et al., 2023) could be complemented with other approaches that account for cumulative damage such as, for example, earthquake and tsunami (Gomez Zapata et al., 2022). We included a sentence in the manuscript to explain this. In the context of Cental Asia, this approach might potentially be applied to earthquake-induced landslides which are common in the region (Saponaro et al., 2014).*

line 392 - typo ("building")
*The typo was corrected*

line 394 - 395: rephrasing necessary to improve readability of sentence
*The sentence was rephrased to explain what we meant. The 2080 layers presented here offer a starting point for  the definition of risk mitigation strategies both at regional and national scale. For example, they can help identifying the typologies that are more prone to generate losses and/or to generate financial risk. Under these considerations, they might be replaced in the future with less vulnerable residential building*

*typologies, as envisaged by many expert and practitioners in the region during exposure development workshops (Peresan et al., 2023).*

line 396 - typo ("top")
*The typo was corrected*

Line 412 - Authors mention the "strong urbanization" but in the results this specific aspect is not shown and discussed in enough detail (see comments above).
*We included a new table that shows the increase of urban areas in each country of Central Asia under the three considered SSPs. We also include a sentence that explains that the urbanization is already evident in Central Asia and has started in the 2000s with an average cities growth rate of 9 to 11% (UNESCAP, 2013). We also added a sentence in the discussion explaining that here we did not consider the interaction between population, urbanization and GPD, but that future work should explore the interplay between the population change and the urbanization process and how they affect exposure and risk indicators, which addresses one of the general comments of the reviewer.*

---

## Author Comment (AC2)

ANSWERS TO REVIEWER 2

This manuscript proposes a regional exposure database featuring the population and residential building stock for the countries of Central Asia. This new dataset has a regional scale, with the appropriate resolution to be used in multi-hazard and risk assessment. Moreover, the authors provide estimates for exposure for the year 2080 that can support current and future risk mitigation strategies. To achieve this, the authors use high resolution population and building datasets to generate gridded exposure datasets. The introduction clearly states the research goals and main novelties. The methodological aspects of the research are very interesting. The authors rely on local data and expert knowledge to provide reliable building characterization and replacement costs. The manuscript presents relevant information for other risk scientists. This information is clearly laid out in maps, figures, and tables, making the paper useful, appealing, and easy to read. The exposure layers for 2080 are a significant addition to the research outputs. These rely on future scenarios of population and urbanization to propose different possibilities for future exposure in Central Asia. The results sec on is brief, clear, and very well written. I praise the authors for presenting the results of their work so well. The manuscript contains a couple of typos, which can be easily corrected, but it is also missing several references, some of which are essential to the research. I strongly suggest thorough proof-read by the authors. Regarding the methodological aspects of the research, there are only a couple of points that need to be explicit in the manuscript. One is regarding the SSPs, specifically the motivation for the scenarios chosen and the uncertain es that they account for in this research (i.e., it seems that future population and urbanization are being accounted for, but not future sustainability, resilience, or vulnerability of the residential building stock). The other point that needs to be clearly addressed is the expected reduction in the future number of buildings, which is most likely due to the fact the future abandoned or unoccupied buildings are not being accounted for in the 2080 exposure dataset. Based on this assessment, I believe the manuscript would be an excellent contribution to the scientific community, and that its main contents are up to the standards of NHESS. I also want to extend my congratulations to the authors for their hard work. My recommendation is to accept the manuscript, subject to minor revisions. Please find my comments line to line below, which I hope can help the authors greatly improve the quality of the final publication.

*Thank you very much for your suggestions. We replied to all comments, included the missing references and proof-checked the document. We added more details on how the future exposure projections were developed, the choice of SSPs and how the progressive replacement of deprecated building typologies was accounted for. We also included new figures that show the exposure layers developed for different building typologies. Please find attached our responses to each comment.*

**Introduction**:
Lines 30-35: Reference Yu et al. 2019 is missing in the references section. The authors need to include it.
*The reference was included, thank you.*

Lines 45-50: Reference UNECE 2017 is missing in the references section. The authors need to include it.
*The reference was included, thank you.*

Lines 55-60: The authors mention that the only regional-scale exposure dataset of residential buildings available at the time (April 2023) is provided by Piiiore et al., (2020). I believe the Global Earthquake Model Foundation has provided an updated exposure model for Central Asia in terms of population, building counts and replacement costs (See Yepes-Estrada et al. 2023 and https://github.com/gem/global_exposure_model). The authors should consider mentioning the update as well.
*The reference was now included, thank you for pointing it out.*

Lines 75-80: The authors state that the first regional-scale exposure dataset for Central Asia was developed by Pittore et al., (2020) using ground-based and remote sensing datasets. The Global Earthquake Model Foundation also has updated exposure datasets covering Central Asia using a bottom-up approach (Yepes-Estrada et al. 2023). Herein, however, the authors maintain that they are proposing the 'first regionally-consistent exposure database for Central Asia'. Were the first efforts and methodologies to map the residential exposure in the region inconsistent? How so? There are novelties in the new dataset the authors propose, but this assertion needs further explanation to be included in the manuscript.

*With respect to the work of Pittore et al. (2020), we update the model using recent country-based data and we use a constant spatial grid of higher resolution (500m) which covers the entire Central Asia region. Also with respect to Yepes-Estrada et al. (2023), we use a higher resolution grid (500m with respect to approximately 30km). Our resolution is, in particular, higher in rural areas, and supports the impact and risk assessment not only for earthquakes but also for floods, under certain limitations that are discussed in the manuscript. We therefore included the Yepes-Estrada work in the references and modified the wording explaining that this is not the first exposure model produced for Central Asia but is the first high-resolution exposure dataset that provides exposure to earthquakes, floods and landslides at a constant spatial resolution of 500m.*

**Methodology**

Lines 120-125: The authors mention that comparing Facebook data versus national census data results in differences in the total population that exceed 20% in 7 oblasts. What is the total number of oblasts under study? What is the difference between the total population estimates from the census and Facebook? Would it be possible to elaborate? The use of Facebook data is an interesting approach and I think it would be interesting for other researchers to see how it compares to national databases.

*At the time of the analysis, Facebook data had been recently released (June 2019) and provided data at a higher resolution (30m) with respect to other datasets such as WorldPop. We compared it with the latest available population census from the year 2021 for Uzbekistan, 2020 for Kazakhstan and Kyrgyz Republic and 2019 for Turkmenistan. Population data was retrieved for 2018 in Tajikistan. However, this data was older than the Facebook dataset and was available only for selected towns and cities, so we did not correct the Facebook layer assuming it to be more reliable. The comparison showed that at the regional scale, the Facebook dataset contains a 5% less population than the sum of the national scale census retrieved. At the national scale, the population in the Facebook dataset was also consistently lower than in national census, with a difference of 1.5, 4, 5 and 8% respectively for Kazakhstan, Uzbekistan, Turkmenistan and Kyrgyz Republic. We noticed that larger discrepancies were associated with the presence of older census data (e.g. for Turkmenistan) while smaller differences are found in Kazakhstan, and Uzbekistan. Large differences are found between the Facebook layer and the national census for Kyrgyz Republic despite the fact that the census was relatively recent. We included more details on the database and its usage, including the total number of Oblasts used for the analysis and the overall difference between the two datasets at regional and national scale and the differences observed between the two datasets.*

Lines 135-140: The Global Earthquake Model building taxonomy by Brzev et al., 2013 has been updated by Silva et al., 2022 (https://doi.org/10.1007/s13753-022-00400-x.).
*The reference was updated, thank you.*

Lines 160-165: The authors used the Facebook population data and adjusted it to respect local estimates of the population. Then the buildings were taken from Pittore et al. (2020). Both datasets end up distributed in variable-resolution grids. These disaggregation techniques can result in an inconsistent number of occupants per dwelling and occupants per building in each cell, especially if datasets developed independently are being combined. Do the time stamps of the population and building datasets agree with each other? (i.e. the population of 2020 and the number of buildings by 2020). Was the final number of occupants per dwelling and building consistent with the subnational local estimates after the distribution? It would be a great addition to the methodology section to briefly mention the checks performed after the disaggregation methods.

*Overall, the timestamps between the dataset are not always matching, which can introduce discrepancies. Regarding the population, we only used recent census data to correct the Facebook dataset, which is dated 2019. In particular, the census dates were 2021 for Uzbekistan, 2020 for Kazakhstan and Kyrgyz Republic, 2019 for Turkmenistan. For Tajikistan, we did not use the census because it was from 2018. Here, we maintained the Facebook constant grid. Since the population datasets were all recent and of similar age, we performed a simple check to make sure that the population never had negative values.*

*As for the buildings data, we updated the Pittore et al. (2020) dataset (that uses buildings data from the time period 2012-2016 and population ancillary data referred to 2015) using country-based data for Kazakhstan and Uzbekistan, referred to 2020. This was done maintaining the same resolution of the original layer. Finally, we disaggregated the buildings layer on the regular grid of the population dataset.*

*The larger discrepancies are expected, in principle, for the countries for which the Pittore et al., dataset was not updated, i.e. Kyrgyz Republic, Tajikistan and Turkmenistan. In order to make sure that there were not discrepancies between the two datasets, we performed the following simple checks:*
*- We made sure that no points were associated with null population and not-null number of buildings, or vice-versa*
*- We checked that the average number of occupants per building (i.e. the fraction between population and number of buildings in each point of the grid) was consistent with the average occupancy defined for Central Asia typologies (Table 1 of the manuscript). No points had an average number of occupants per building larger than 200, which would have required additional checks.*

*A sentence on this was added to the manuscript, underlining the importance of tackling temporal variability of exposure and checking that the results of the disaggregation are realistic and consistent among the variables which have known relation (e.g. building occupancy and population).*

**Development of exposure layers for 2080**
Lines 210-2020: HAZUS is mentioned throughout the manuscript, but I could not find the reference in the reference section (HAZUS 2021? FEMA 2021? Inventory technical manual?). This is the third reference that is either missing from the manuscript or difficult to find in the corresponding section. There is also a typo: 'manual (2021).g dife Costs'. I strongly recommend the authors to proofread the manuscript carefully.
*Thank you, the Hazus reference has been included, we referred to the HAZUS inventory technical manual, available at: https://www.fema.gov/sites/default/files/documents/fema_hazus-inventory-technical-manual-4.2.3.pdf, accessed 20/10/2021, edited by FEMA. We also carefully proofchecked the manuscript and apologize for the typos and flaws.*

Lines 235-245: The SSPs chosen to develop these layers were SSP1, SSP4 and SSP5. This is a very interesting choice. However, the authors mention that these scenarios envisage different development drivers, but provide no further motivation for selecting these. For example, why was the SSP2 not included? The 'middle of the road' considers less strong deviations from the current fertility trajectories, hence it is the likely scenario in terms of the future population and urbanization. Could the authors elaborate on what motivated this choice? Was supporting risk management strategies a part of this motivation? How so?
*The number of SSPs to be considered in the analysis was limited to three by the need to perform subsequent risk analysis which were time consuming for other research partners (Salgado et al., and Berny et al., this volume). We decided to use three 'extreme' SSPs in order to highlight the larger variations expected, and to create an upper and lower bound for expected exposure changes with respect to more 'middle of the road' scenarios. The choice of SSP1 was also motivated by the willingness to highlight the role of governance and international cooperation and show how it affects exposure. However, we agree with the reviewer that other scenarios such as SSPs are very relevant (and could represent a more likely future pathway). We included a sentence in the manuscript to explain this and open to future work that analyzes in more detail the implications of different SSPs for exposure assessment and disaster risk reduction.*

Lines 245-265: It is unclear how the SSPs are used beyond the scenarios of future population and urbanization. The uncertainty regarding the future population is taken from the SSPs. The different future population scenarios are used to infer a future number of buildings, and the urbanization layers inform location and density. However, there seems to be no uncertainty considered in the future characterization of the buildings, which influences the final vulnerability and sustainability of the future exposure datasets. This would be a very difficult task, and the authors rightly rely on expert judgement alone to propose a single set of rules for future building characterization. There is no mention of this set of rules changing depending on the SSPs. If this is the case, the authors need to be explicitly clear in the manuscript that the SSPs only inform the population figures and building allocation. This is important given the choice of SSPs by the authors. For example, the SSP1 (sustainability with low emissions) and the SSP4 (high inequality with high emissions) are clearly opposite scenarios. For example, a reader could understand that sustainability and a more resilient built environment were at the core of creating the future building characterization for an SSP1 scenario, but it doesn't seem to be the case. I think this section really needs more clarity on this point.
*Thank you for your comment. We agree with you that, right now, it is somehow unclear how SSPs influence the assessment of future exposure. We explicitly say that SSPs are used only to inform on population variation and residential buildings allocation, and that conversion rules defined with local experts are*

*assuemed to be the same in all SSPs. We also included a few sentences to explain the limitations of the approach, underlining the need for performing more sophisticated analysis in the future where scenarios are based not only on the number of populations and buildings, but also on the different economic system envisaged in each SSP and the socio-economic consequences of its adoption. I think that implementing an analysis of this kind would probably lead to higher discrepancies in the exposure layers, highlighting the beneficial or negative impact of the economic model on the overall exposure to disasters.*

**Results**

The results section is brief, clear, and very well written. I praise the authors for presenting the results of their work so well. I would even recommend the authors include a figure like Figure 3 for all the sub typologies but only if it is possible within their time constraints.

*We generated new plots for Figure 3, which now shows the spatial distribution of three selected sub-typologies (URM1, RC1 and RCPC1) at the regional scale and for a selected area.*
*Figures are found below:*

[Figure]

[Figure]

The only comment here is regarding Table 5. The building variation is mostly negative in the scenarios, given the population prospects of Central Asia. However, it was not explained in the methodology why the building count is decreasing. I believe that it is because the dataset is meant to include occupied structures only and abandoned or unoccupied structures are not being considered. This needs to be explicit in the methodology.

This is important because a sustained population decrease does not always lead to a sustained decrease in the exposed number of structures. For example, despite a steady decline in population, the number of total dwellings exposed in Albania rose by more than 25% between 2001 and 2011 (www.instat.gov.al). In other words, residential exposure in Albania (new dwellings become new buildings) is increasing despite the decreasing population, as there are other factors beyond population driving changes in residential exposure. Moreover, less population might lead to fewer occupied structures, but abandoned or unoccupied structures are still at risk of flooding, shaking, and causing debris, which will incur an economic loss for society. If the datasets the authors proposed account only for occupied structures, that is perfectly reasonable, but the limitations of this approach should be mentioned as well.

*Thank you for the comment, we clarified the methodology explaining in a more explicit way how unoccupied structures are considered in the analysis. In our work, we distinguish between urban and rural areas. In urban or expected-to-be-urban areas, we only account for occupied structures, and we replace old structures with the new typologies according to set of rules previously defined. In rural areas the deprecated building types are maintained in the 2080 exposure layer in order to avoid underestimating the risk related to weak typologies which might still be in use, or not demolished, despite their age. In urban areas, abandoned or unoccupied structures are not being considered as they are assumed to be demolished during the process of building new structures. However, we agree with you that this need to be clear as it might lead to underestimate the overall impact (which might include unoccupied buildings as well). This was clarified in the text.*

**Discussion**

Lines 345-350: The authors mention that the final data is provided in GED4ALL format. Is it open access already? Consider using 'will be provided'.

*At the time of the submission, the link for data sharing was not active. We included it in the data availability statement together with the license under which data are made available.*

**References**

Please include the missing references

*The missing references were added, thank you.*

---

## Author Response (AR2)

NHESS-2023-94 Author's response

Dear editor,

please find attached the manuscript with the required changes. The list of references was reviewed and Fig. 3 was modified using a color scale that ensure readers with color vision deficiencies to interpret the findings.

Thank you very much,

Chiara Scaini, on behalf of the co-authors.